



# Optimized step size control within the Rosenbrock solvers for stiff chemical ODE systems in KPP version 2.2.3_rs4

Raphael Dreger[1], Timo Kirfel[1,4], Andrea Pozzer[2], Simon Rosanka[3], Rolf Sander[2], and Domenico Taraborrelli[1,4]

[1]Forschungszentrum Jülich GmbH, Institute of Climate and Energy Systems, ICE-3: Troposphere, Jülich, Germany
[2]Atmospheric Chemistry Department, Max Planck Institute for Chemistry, Mainz, Germany
[3]Department of Chemistry, University of California, Irvine, California, United States
[4]Centre for Advanced Simulation and Analytics (CASA), Forschungszentrum Jülich, Jülich, Germany

**Correspondence:** Raphael Dreger (r.dreger@fz-juelich.de) and Domenico Taraborrelli (d.taraborrelli@fz-juelich.de)

**Abstract.**

Numerical integration of multiphase chemical kinetics in atmospheric models is challenging. The underlying system of ordinary differential equations (ODEs) is stiff and thus difficult to solve. Rosenbrock solvers are a popular choice for such tasks. These solvers provide the desired stability and accuracy of results at an affordable yet large computational cost. The latter is crucially dependent on the efficiency of the step size control. Our analysis indicates that the local error, which is the key factor for the step size selection, is often overestimated, leading to very small sub-steps. In this study, we optimized the first-order step size controller most commonly employed in Rosenbrock solvers. Furthermore, we compared its efficiency to a second-order step size controller. We assessed the performance of the controllers in both a box and a global model for very stiff ODEs. Significant reductions of the computation time were accomplished with only marginal deviations in the results compared to the standard first-order controller. This was achieved not only for gas-phase chemistry but also for the more complex aqueous-phase chemistry in cloud droplets and deliquescent aerosols. Depending on the selected chemical mechanism, significant improvements were already achieved by simply adjusting heuristic parameters of the default controller. However, especially for the global model, the best results were achieved with the second-order controller, which reduced the number of function evaluations by $43\%$, $27\%$ and $13\%$ for gas-phase, cloud and aerosol chemistry, respectively. The overall computational time was reduced by over $11\%$ while requiring only minimal adjustments to the original code. Analysis of a 1-year integration period showed that with the second-order controller, the deviations from the reference simulation stays below $1\%$ for the main tropospheric oxidants. The results presented here show the possibility of more efficient atmospheric chemistry simulations without compromising accuracy.

## 1 Introduction

Atmospheric chemistry modeling plays an essential part in understanding and predicting air composition and its interactions within the Earth's system. The chemical kinetics used in these models can be very complex (e.g. Rosanka et al., 2021a; Pozzer





et al., 2022) and rely on the numerical solution of stiff systems of ordinary differential equations (ODEs), normally responsible for largest part of the computing time (Christou et al., 2016). The systems of ODEs can be written as

$$y' = P(t,y) - L(t,y) \tag{1}$$

where $y$ is the array of concentrations (in $\mathrm{cm}^{-3}$), $y'$ is derivative of $y$ with respect to time $t$, $P(t,y)$ and $L(t,y)$ are the production and loss terms of the species, respectively. For stiff ODEs, implicit solvers with adaptive time stepping are preferably used (Sandu et al., 1997b). Stiffness refers to a situation in which an explicit ODE solver needs a "too small" time step and thus too many discretization steps to approximate the solution at the desired precision. However, a more objective definition of stiffness exists and a criterion for identifying a problem as stiff has been defined by Spijker (1996):

$$\max Re(\lambda_j) < 0, \; \frac{\max|Re(\lambda_j)|}{\min|Re(\lambda_j)|} >> 1 \tag{2}$$

where $Re(\lambda_j)$ are the real parts of the eigenvalues of the Jacobian matrix, $J(t,y)$, associated with the ODE system from Eq. (1). Atmospheric chemical kinetic models satisfy this criterion when the mechanism contains very fast as well as very slow reactions (Sandu et al., 1997b). In the typical operator splitting framework the chemical kinetics has to be solved every few minutes after many non-chemical processes, e.g. emissions, which have perturbed the chemical composition. Thus, numerical integrators are not efficient because at every start, for a model time step of 10-20 minutes, they require very short initial sub-steps ($10^{-5}$ s or shorter). In chemical kinetic terms, this is generally related to the perturbation of the chemical environment (the long-lived species) that determines the concentrations of the short-lived lived species. In mathematical terms, stiffness is related to the large negative values of $\lambda_j$ that are associated to the diagonal elements of the Jacobian matrix ($J_{jj}(t,y)$). The latter are usually related to the loss term $L_j(t,y)$ and thus to the lifetime of the species $j$ (Turanyi et al., 1993). However, this is not the case for multiphase chemical mechanisms, for which large negative values of $\lambda_j$ cannot always be associated to a single short-lived species but rather to fast acid-base equilibria or phase-transfer reactions (Sandu et al., 1997b). Especially stiff are the ODE systems describing chemical kinetics in deliquescent aerosols with low liquid water content (LWC) and extremely fast outgassing of dissolved species. This has strongly limited the ability of performing multiphase chemistry simulations with explicit kinetics (Kerkweg et al., 2007). Only recently it has been possible to perform numerical integration of chemical kinetics in deliquescent aerosols throughout the troposphere in a global model for LWC as low as $10^{-14}$ $\mathrm{m}^3(\mathrm{aq})\,\mathrm{m}^{-3}(\mathrm{air})$ (Rosanka et al., 2024). However, such simulations pose a large burden on the computational resources needed and limit the understanding of the role of multiphase chemistry for atmospheric composition. Thus, there is a need for more efficient ODE integrators. Many numerical integrators are available and have been used for atmospheric applications (Zhang et al., 2011). At the maximal error that is usually accepted (1 %), Rosenbrock methods have proven to be the most efficient among the implicit solvers that offer the required stability for stiff problems (Sandu et al., 1997a). Linearly-implicit Rosenbrock methods are a popular choice for solving such stiff ODE systems because of their suiting stability while not relying on costly solutions of nonlinear equations (Zhang et al., 2011). However, the precision and efficiency of these discretization methods depend on





the control strategy used to adaptively adjust the integration step size. Ways of making these integrators more efficient have involved for instance manual or semi-automatic reduction of the ODE system size (Sander et al., 2019; Wiser et al., 2023; Lin et al., 2023). What is less explored are the improvements in efficiency by using a more sophisticated step size control. The adaptive time step controller that is usually employed in many applications, including chemical solvers, controls the locally produced error $r_i$ to be within a tolerable threshold $\varepsilon$. The asymptotic behavior of the numerical solver is used to adaptively approximate the largest possible step size that is within the tolerance. The resulting step size controller

$$h_{i+1} = h_i \sqrt[k]{\frac{\varepsilon}{r_i}} \tag{3}$$

was introduced by Hairer et al. (1993), where $h_i$ is the time step size for the $i^{th}$ step and $k$ is usually the order of convergence of the solver plus one ($k = p+1$). $\varepsilon$ is a predefined relative error tolerance and $r_i$ is the local error produced by the solver's solution $y_i$ compared to the real solution $y$ at the corresponding point in time $t_i$. Chemical solvers usually make use of low values of $p$ (Shampine and Witt, 1995; Sandu et al., 1997b). Additionally, Rosenbrock solvers, as part of the Runge-Kutta solver family, offer an efficient approximation of the local error with the help of a second embedded solver which yields a second solution $\hat{y}$ of lower order $\hat{p}$ with nearly no extra effort (Hairer et al., 1993). This approximation will be denoted as $l_i = y_i - \hat{y}_i$ from now on and is a substitute for the exact local error $r_i$. In practice, this controller is extended with some heuristic measures to ensure stability. Equation (4) shows the heuristic extensions and represents the currently used controller used in our applications that we investigate in this work:

$$h_{i+1} = h_i \cdot \min\left(q_{max}, \max\left(q_{min}, \delta \sqrt[\hat{p}+1]{\frac{1}{||l_i||}}\right)\right) \tag{4}$$

The new step size gets multiplied with a safety factor $\delta$ and the step size growth is limited with an upper and a lower value, called $q_{min}$ and $q_{max}$ respectively. This practice was also proposed by Hairer et al. (1993) and also deemed appropriate for Rosenbrock solvers in Hairer and Wanner (1996). Nevertheless, a small difference to the implementation we use is that the $\varepsilon$ in the numerator is replaced by one because the tolerance is incorporated into the local error norm $||l_i||$. Equation (5) shows the implemented error norm $||l_i||$, which is similar to the scaled Euclidean norm from Hairer et al. (1993). There $y_{i,j}$ is the $j^{th}$ component of the solver solution for the $i^{th}$ time step and $\hat{y}_{i,j}$ the corresponding solution of the embedded, lower order integrator. The scaling factor of the difference contains the absolute and relative tolerances, $\varepsilon_{abs}$ and $\varepsilon_{rel}$, which have to be specified by the user. This normalizes the error, which means that steps get accepted if the error norm is equal to or less than one. In the context of atmospheric chemistry, rather large tolerances as high as $\varepsilon_{rel} = 10^{-2}$ and $\varepsilon_{abs} = 100\,\mathrm{cm}^{-3}$ are used (Zhang et al., 2011) because other uncertainties in Earth System models are considered to be larger.

$$||l_i|| = \sqrt{\frac{1}{n} \sum_{j=1}^{n} \left(\frac{y_{i,j} - \hat{y}_{i,j}}{\varepsilon_{abs} + \varepsilon_{rel} \cdot \max(|y_{i-1,j}|, |y_{i,j}|)}\right)^2} \tag{5}$$

However, Söderlind (2003) developed a suite of time-step controllers based on control theory aspects. This idea is practicable because the aspired relative tolerance given by the user can be interpreted as a target value and the local error in each step as



the actual value that the controller needs to adjust by reacting to the behavior of the ODE system. Many of these controllers are of higher order of adaptivity than the first-order controller presented in Eq. (4). One of the second-order controllers is the
H211b

$$h_{i+1} = h_i \left( \frac{\varepsilon}{||l_i||} \right)^{1/(b \cdot k)} \left( \frac{\varepsilon}{||l_{i-1}||} \right)^{1/(b \cdot k)} \left( \frac{h_i}{h_{i-1}} \right)^{-1/b} \tag{6}$$

with $b$ and $k$ being free parameters of the controller. The parameter $k$ is the root exponent that is already known from the currently used controller, where it was set to $\hat{p}+1$. This means that one could set $k = \hat{p}+1$. However, Söderlind (2003) hints that this selection is in no way predetermined. To be more precise, the value should not be lower than $\frac{\hat{p}+1}{2}$ and not "too large", $\hat{p}+1$
is just a value that always produces a converging step size control (Deuflhard and Bornemann, 2008). The second parameter should be $b > 1$ for stability, with increasing values creating a smoother and robuster step size sequence.

In this article, we improve the step size control implemented in the Rosenbrock solvers available with the Kinetic Pre-Processor (KPP, version 2.2.3, Sandu et al. (1997a); Sandu and Sander (2006)). The KPP software is mainly used to solve the
very stiff sets of ODEs resulting from the kinetics of atmospheric chemical mechanisms of varying complexity in the gas phase (e.g., Pozzer et al., 2022), in cloud droplets (e.g., Rosanka et al., 2021a) and deliquescent aerosols (Rosanka et al., 2024). In brief, KPP translates input files containing chemical reactions and rate constants into source code files containing the resulting ODE system, as well as an integrator chosen by the user. KPP is frequently applied in atmospheric models across multiple scales, ranging from simple box models (e.g., CAABA by Sander et al., 2019) to complex regional and global atmospheric
chemistry models (e.g., EMAC by Jöckel et al., 2010). We first evaluate the step size control within the CAABA box model. Based on the results, we present adjustments to the first-order controller and compare its performance to the second-order step size controller H211b presented by Söderlind (2003). To evaluate our adjustments under varying atmospheric conditions, we also performed global simulations with EMAC.

## 2 Methodology

### 2.1 Chemical kinetic model

Multiphase chemistry is represented by the kinetic model in MECCA ("Module Efficiently Calculating the Chemistry of the Atmosphere") (Sander et al., 2019). The rate of phase-transfer reactions is governed by liquid water content (LWC), water solubility and mass-transfer coefficient according to Kerkweg et al. (2007) following the formulation by Sander (1999). The inorganic gas-phase chemistry mainly follows the recommendations by JPL (Burkholder and et al., 2019) and IUPAC (Walling-
ton et al., 2018). The organic gas-phase chemistry in MECCA is represented by the Mainz Organic Mechanism (MOM), which includes organics up to $C_{10}$-molecules for isoprene (Taraborrelli et al., 2012; Nölscher et al., 2014; Novelli et al., 2020), monoterpenes (Hens et al., 2014; Mallik et al., 2018) and aromatics (Cabrera-Perez et al., 2016; Taraborrelli et al., 2021). In total, MOM represents 735 chemical species and 2196 reactions. The inorganic aqueous-phase chemistry is represented by a comprehensive set of reactions for species containing oxygen, nitrogen, sulfur and halogens (Kerkweg et al., 2008). The





**Table 1.** Overview over the used (sub)models and the corresponding specifications of the chemistry and integrator for the performed simulations.

| model/submodel | phase | $\mathbf{LWC}_{min}$ $(\mathrm{m}^3(\mathrm{aq})\,\mathrm{m}^{-3}(\mathrm{air}))$ | $\mathbf{radius}_{min}$ (m) | integrator | $\varepsilon_{rel}$ | $\varepsilon_{abs}$ $(\mathrm{molec\,cm}^{-3})$ |
|---|---|---|---|---|---|---|
| CAABA/MECCA | gas+aqueous | $1.08 \cdot 10^{-12}$ | $8.82 \cdot 10^{-8}$ | Ros3 | $10^{-2}$ | 1 |
| EMAC/MECCA | gas | - | - | Ros3 | $10^{-2}$ | 10 |
| EMAC/SCAV | aqueous | $1 \cdot 10^{-9}$ | - | Rodas3 | $10^{-2}$ | 10 |
| EMAC/GMXe-AERCHEM | aqueous | $5 \cdot 10^{-13}$ | $1 \cdot 10^{-8}$ | Rodas3 | $10^{-3}$ | 1 |

aqueous-phase chemistry of organics up to $C_4$-molecules is represented by the Jülich Aqueous-phase Mechanism of Organic Chemistry (JAMOC, Rosanka et al., 2021b). JAMOC represents 792 chemical species and 1148 reactions (Rosanka et al., 2021b).

## 2.2   Box model

We used the community box model CAABA ("Chemistry As A Boxmodel Application") in this study (Sander et al., 2019).

It represents chemical and physical (emission, dry deposition, entrainment, photolysis) processes in the atmosphere in a simplified manner. CAABA is coupled to the submodel MECCA, which contains a comprehensive set of multiphase chemical reactions. Based on these, KPP creates one large system of ODEs for the gas phase and two condensed phases, representing populations of deliquescent aerosols (LWC=$1.08 \cdot 10^{-12}$ $\mathrm{m}^3(\mathrm{aq})\,\mathrm{m}^{-3}(\mathrm{air})$, $r = 8.82\ 10^{-8}$ $m$) and cloud droplets (LWC=$3.04 \cdot 10^{-11}$ $\mathrm{m}^3(\mathrm{aq})\,\mathrm{m}^{-3}(\mathrm{air})$, $r = 1.67\ 10^{-6}$ $m$). The Rosenbrock integrator used is Ros3 with relative and absolute

tolerances $\varepsilon_{rel} = 10^{-2}$ and $\varepsilon_{abs} = 1\,\mathrm{cm}^{-3}$, respectively. We have defined three scenarios which emphasize different parts of the chemical mechanism:

1. The marine boundary layer (MARINE): Gas-phase chemistry interacts with aqueous-phase chemistry in marine aerosol particles.

2. A coastal megacity (MEGACITY): Anthropogenic emissions of hydrocarbons interact with halogen- and sulfur-containing

compounds from the sea (Crippa et al., 2018; Huang et al., 2015).

3. A pristine tropical rain forest (FOREST): Biogenic isoprene and terpenes are emitted into the air and then oxidized in a complex degradation scheme.

## 2.3   Global model

The ECHAM/MESSy Atmospheric Chemistry (EMAC) model is a numerical chemistry and climate simulation system that in-

cludes submodels describing tropospheric and middle atmosphere processes and their interaction with oceans, land and human



influences (Jöckel et al., 2010). It uses the second version of the Modular Earth Submodel System (MESSy2) to link multi-institutional computer codes. The core atmospheric model is the 5th generation European Centre Hamburg general circulation model (ECHAM5, Roeckner et al., 2006). The physics subroutines of the original ECHAM code have been modularized and reimplemented as MESSy submodels and have continuously been further developed. Only the spectral transform core, the flux-140 form semi-Lagrangian large scale advection scheme and the nudging routines for Newtonian relaxation are remaining from ECHAM. For the present study we applied EMAC (MESSy version 2.55.2) in the T42L31-resolution, i.e. with a spherical truncation of T42 (corresponding to a quadratic Gaussian grid of approx. 2.8 by 2.8 degrees in latitude and longitude) and with 31 vertical hybrid pressure levels up to 10 hPa. The global model is run in the Quasi Chemical Transport Mode (QCTM) without feedbacks between chemistry and physics (Deckert et al., 2011). This ensures that the meteorology of the model and its 145 influence on tracer concentration, remain binary identical in all the simulations performed. Therefore, any difference in tracer abundance is purely attributable to the different integration of the chemical ODE.

Like CAABA, EMAC uses MECCA to represent chemical kinetics. The same parameter space of the box model can be explored for testing in the global model. Unlike the box model and early EMAC simulations (Kerkweg et al., 2007), EMAC now relies on so-called operator splitting in representing chemistry in the gas phase and in deliquescent aerosols separately, in 150 series (Rosanka et al., 2024, their Fig. 1a). In the simulations performed in this study, we use MECCA to represent gas-phase chemistry including some heterogeneous reactions, whereas aqueous-phase chemistry in convective and large-scale clouds and rain is represented using the SCAV (SCAvenging) submodel (Tost et al., 2006). Aerosol processes are represented using the MESSy's Global Modal-aerosol eXtension (GMXE, Pringle et al., 2010) submodel. Here, aqueous-phase chemistry in accumulation and coarse deliquescent aerosols is represented by the sub-submodel AERosol CHEMistry (AERCHEM, Rosanka et al., 155 2024), which is part of the GMXe submodel. AERCHEM is executed only when the liquid water content of the aerosols is larger than $5 \cdot 10^{-13} \, \mathrm{m^3(aq) \, m^{-3}(air)}$. From the available KPP Rosenbrock methods, we chose Ros3 for MECCA (gas phase) and Rodas3 for SCAV and GMXe-AERCHEM (aqueous phase) due to favorable performance and stability (Rosanka et al., 2024). For the tolerances, MECCA and SCAV both use $\varepsilon_{rel} = 10^{-2}$ and $\varepsilon_{abs} = 10 \, \mathrm{cm^{-3}}$. Due to higher stiffness and lower stability when solving aqueous-phase chemistry in deliquescent aerosols, AERCHEM uses a relative tolerance of $\varepsilon_{rel} = 10^{-3}$ 160 and an absolute tolerance of $\varepsilon_{abs} = 1 \, \mathrm{cm^{-3}}$.

For all EMAC simulations performed in this study, biogenic emissions are represented by the Model of Emissions of Gases and Aerosols from Nature (MEGAN, Guenther et al., 2012). Biomass burning related emissions are calculated by MESSy's BIOBURN submodel, which combines biomass burning emission factors with dry-matter combustion rates obtained from the Global Fire Assimilation System (GFAS) based on the Moderate Resolution Imaging Spectroradiometer (MODIS) satellite 165 instruments (Kaiser et al., 2012). Sea-spray aerosol emissions are calculated online following the methodology by Kerkweg et al. (2006). We represent mineral dust as bulk inert dust; i.e., no crustal elements are emitted, with online emissions calculated following Astitha et al. (2012). All anthropogenic emissions follow the Emissions Database for Global Atmospheric Research (EDGAR v4.3.2; Crippa et al., 2018).





## 2.4 Simulations performed

Global model simulations are much more complex than box model ones and more efficient simulations would save costly computation time on High Performance Computing (HPC) systems, which offers the opportunity for longer simulation periods or more complex chemistry mechanisms. However, we started with CAABA box model simulations because its reduced complexity allows an easier analysis of the step size selection and local error behavior of the step size controllers.

The box model simulations covered a simulation period of 12 hours and output was written at every model time step of 10 minutes. We also added output of the local error, step size and number of function evaluations for every integrator time step. All simulations were done for all three presented scenarios. Besides reference simulations with the default values of the first-order controller, we made simulations for every single parameter change, like an increased safety factor. To evaluate changes in precision, we also made another reference simulation with a fully implicit Radau integrator of order five and a relative tolerance of $\varepsilon_{rel} = 10^{-7}$ and absolute tolerance of $\varepsilon_{abs} = 1\,\mathrm{cm}^{-3}$.

Given that global model simulations require much more computational effort, we reduced the number of values tested for each parameter and the simulation period for the first set of simulations only covers the first day of the year 2009. MECCA, SCAV and AERCHEM produce a single ODE system each. Therefore, in each one-day simulation we changed one parameter of one submodel. Output was produced after each model time step of 15 minutes. After exploring the influence of the different parameters, we also made one-year simulations for the year 2009 with the best performing parameters of the first-order controller and the second-order controller. For run-time and error comparisons, a reference one-year simulation which contains the current default settings was made as well. To have an acceptable amount of I/O, we only wrote output every 23rd hour. With this selection we have output for every day of the year and every hour of the day. The global simulations were done on the JUWELS Cluster Supercomputer from the Research Center in Jülich. We used eight nodes with 48 cores each.

## 3 Step size control in the box model

### 3.1 First-order controller

In this section, the efficiency of the standard first-order step size controller from Eq. (4) is assessed on the basis of the CAABA simulations for the three presented scenarios. Hereby, we focus on two key aspects: (1) the step size and (2) the local error. The specific KPP implementation of the controller and the used error norm are also described in Sandu et al. (1997a). Further, we provide pseudocode of the implementation in Appendix A.

### 3.1.1 Evaluation

The local error estimation is quite crucial for an efficient solution because it is the main factor influencing the next step size. If the estimated value is too large, this could lead to an inefficient step size sequence with unnecessarily small step sizes. For this reason, we compared a more accurate approximation of the local error to the one calculated by the integrator. To get this precise estimation, we proceeded as follows:





1. calculate step size $h$ as usual

2. divide the area $[t, t+h]$ into five equally distant areas

3. each of the five areas is viewed as an own local integration problem, starting at $t + \frac{j}{5}h$ and ending at $t + \frac{j+1}{5}h$, $j \in \{0,1,2,3,4\}$

4. the local problems are solved one after another with the same solver and parameters used for the actual ODE

5. the result of the fifth local problem is a precise approximation of the solution at position $t + h$

6. instead of the embedded solution $\hat{y}$, we can now use this solution to calculate a more precise local error norm $||l||$

The division into five equally distant areas produces a solution with a precision of $\varepsilon_{rel}^5$, which is sufficient for our investigations. This procedure obviously required much more computation time and is not practical in real applications, but provides a very precise local error. For simplification, we refer to this estimation in the following as the real local error.

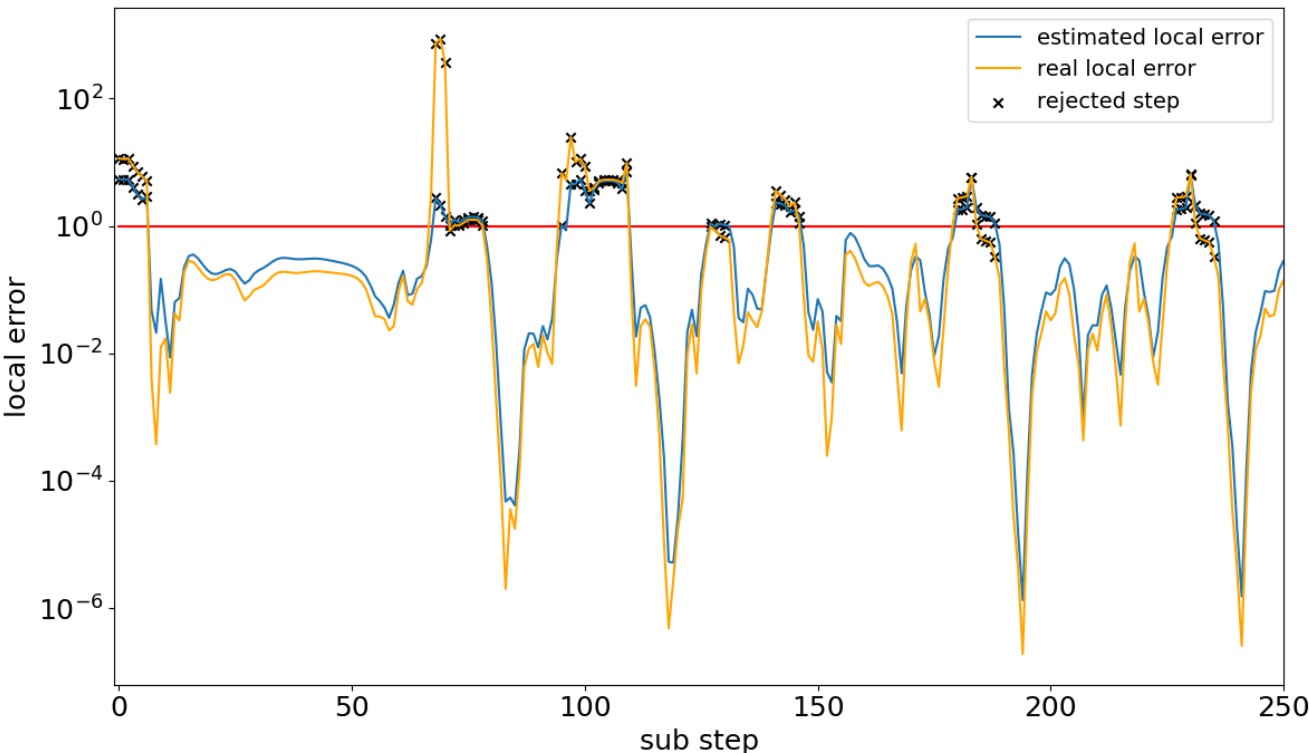

**Figure 1.** Comparison between the estimated local error of the embedded Rosenbrock solver (blue) and the more precise approximation of local error (orange) for the MARINE scenario. The figure shows a nearly constant overestimation of the local error for accepted steps. Also, regular sharp drops in the local error occur. Both findings mark potential for efficiency improvements. The red line marks the acceptance threshold of each sub time step.





Exemplified by the MARINE scenario, Fig. 1 shows that the general course of the estimated local error was fairly good and closely follows the one of the real local error. However, it can be seen that the solver nearly always overestimated the local error for accepted steps. Furthermore, the error frequently became very small, sometimes going below $10^{-6}$ even though the border for acceptance is one. Both of these findings show potential to increase the efficiency of the controller while staying within the desired tolerances. The overestimation leads to smaller step sizes than necessary for the majority of the steps, leading to many

avoidable computations. This is because, by definition of the controller in Eq. (4), $||l_a|| > ||l_b||$ implies $h_a < h_b$. In addition, in some cases the overestimation led to step rejections even though the real error would have been accepted, this happened in less than $5\%$ of the visualized 250 steps. Also, the regularly occurring local error dips indicate that there were areas where the selected step size was much smaller than necessary. Focusing on the local error dips, Fig. 2 illustrates that the drops of the local error occurred in areas where the step size was drastically increasing after the step size dropped to a very small value.

This indicates that the step size could grow faster in these areas than it does currently.

    Generally, our findings indicate that a more aggressive strategy with higher step size selection could work to reduce the required computations while still being within the accepted error margin. To be able to verify this, we compared the simulations with changed parameters to the simulation with the Radau integrator. With these reference results, the number of significant digits places was calculated (see Appendix B). Evaluations of the precision of the standard first-order controller showed that

the component-wise median number of significant digits for the MARINE scenario was up to four, instead of the aspired two (see Table B1). The average was also noticeably above the target of $1\%$.

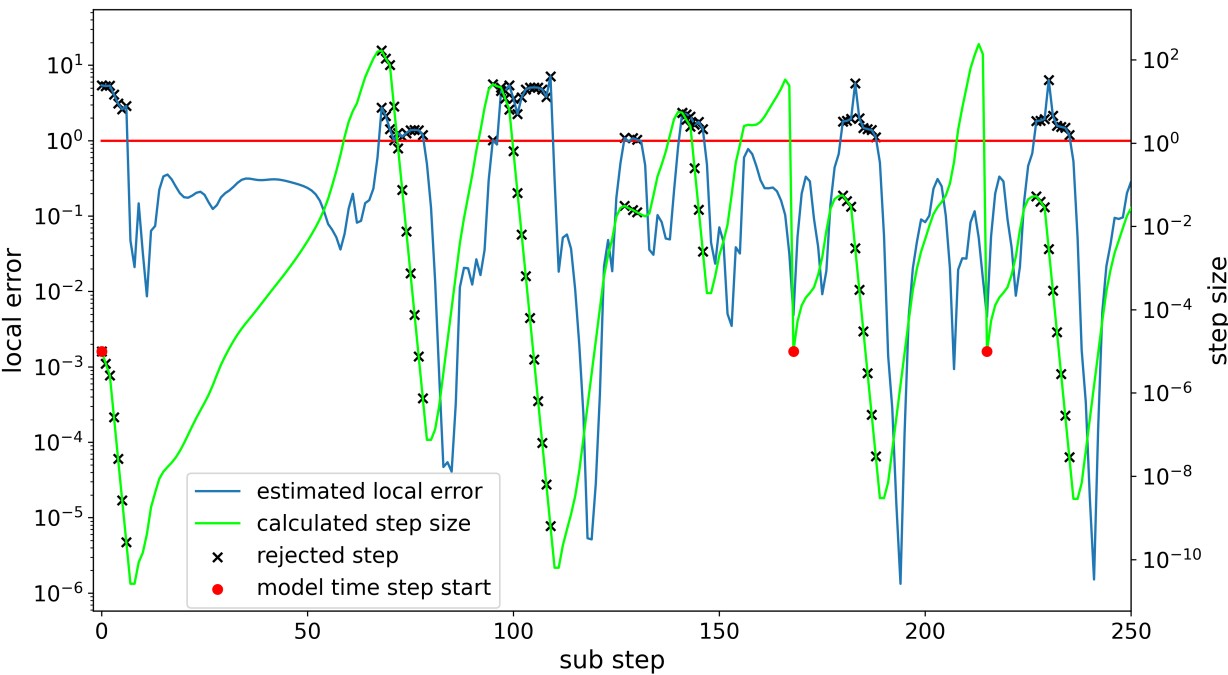

**Figure 2.** The estimated local error (blue) and the selected step size (green) for each substep of the MARINE scenario simulation. The strong local error dips occur when the step size is increasing after a strong decrease. This indicates that faster step size growth may be suitable. Red dots mark the beginning of a model time step. The first model step requires significantly more substeps than the others.

### 3.1.2 Optimization

The investigations from the last section demonstrated that there is considerable potential in improving the step size selection to speed up the integration process without losing the aspired precision. This led to the idea of adjusting the step size controller to

be more generous with the step size in some areas to consider the overestimation of the local error and its regular drop to small orders of magnitude. The easiest way to adjust the behavior of the standard controller were the heuristic parameters listed in Table 2. Most of the default values were already suggested by Hairer et al. (1993) as universally valid options because they are more conservative and focus on stability. Our goal was to find values that suite the ODE systems of the tested mechanisms yielding a lower numerical workload. Therefore, a range of appropriate values was tested for each parameter independently. As

a measure for the computational effort of the integrator, the number of function evaluations was taken, because it is independent of hardware and computations outside the ODE system, in contrast to other measures, e.g. the simulation run time. Optimally, the amount of computations decreases without the single digit accuracy (SDA) dropping below two. SDA represents the number of significant digits.





**Table 2.** Default heuristic parameters of the standard first-order controller as from KPP2 (Sandu and Sander, 2006).

| Parameter | Default Value | Description |
|---|---|---|
| $\delta$ | 0.9 | Safety factor: considers that the local error is just estimated |
| $q_{max}$ | 6 | Upper limit for the growth of the step size |
| $q_{min}$ | 0.2 | Lower limit for the decrease of the step size |
| $r$ | 0.1 | Reduction factor: multiplied with the step size after consecutive rejections |
| $h_{start}$ | $10^{-5}$ | Starting step size |

The historically grown, heuristic assumption to decrease the step size with a safety factor ($\delta < 1$) for precautionary reasons,
is in contrast to the results of our local error analysis. Based on our analysis in Sec. 3.1.1 we now know the actual ratio between
the estimated and the real local error. This made it possible to get a founded estimation of the safety factor. Equation (7) shows
how we estimated the values used for this work.

$$\delta \sqrt[p+1]{\frac{1}{||l||}} \overset{!}{=} \sqrt[p+1]{\frac{1}{\frac{1}{V}||l||}} \tag{7}$$

The left most side of the equation shows the relevant part of the controller Eq. (4). If we assume that the estimated local error
is larger than the real local error by a factor of $V$, we can incorporate this factor into the controller equation by using $\frac{1}{V}||l||$ as
denominator. By factoring $V$ out of the root in Equation (7), we get $\sqrt[p+1]{V}$ as estimation for a new safety factor.

$$\delta = \sqrt[p+1]{V} \tag{8}$$

This results in a safety factor of $\delta \approx 1.4$, slightly varying for each CAABA scenario. Simulations with various values for
$\delta \in [0.9, 1.7]$ showed that values near the calculated optimum led to a quite significant reduction in function evaluations without
a meaningful decrease of precision, as shown in Fig. 3 for the MEGACITY and MARINE scenario. The figures display work-
precision diagrams similarly to Sandu et al. (1997b). Instead of the CPU time, the number of function evaluations is plotted
against the SDA of the least precise component with the largest relative error. Details on how SDA is calculated can be found in
Appendix B. Overall, values in the range of $\delta \in [1.3, 1.5]$ seemed to be the most promising because they provide a significant
reduction in function evaluations of up to $31\%$ while ensuring a smoother step size sequence than higher values. The most
notable finding was that the first increases have a big impact on the reduction of work, but the bigger the safety factor got, the
smaller the influence became.

The analysis of the local error led to the assumption that the step size may grow faster in certain situations to address the
significant drops in the local error. Therefore, increasing the upper growth limit factor $q_{max}$ appeared as a fitting measure
to reduce the numerical burden. Figure 4 seems to support this theory. It shows the calculated growth factor and the upper
limit $q_{max}$ that clips exceeding values. The results were taken from the MARINE scenario simulation. There seems to be a





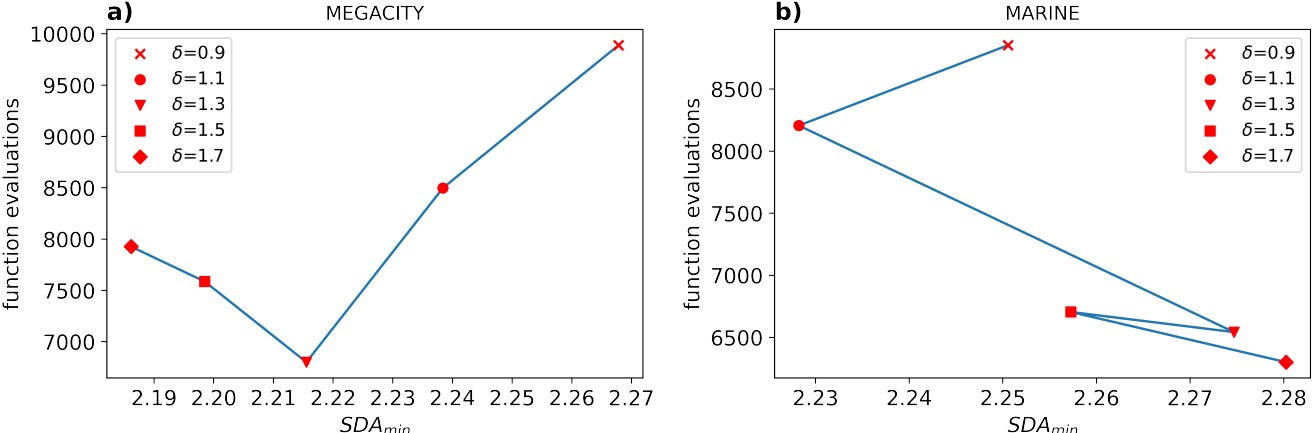

**Figure 3.** Work-precision diagrams for **a)** the MEGACITY scenario and **b)** the MARINE scenario. Generally, a larger safety factor $\delta$ produces less function calls. For MEGACITY the SDA decreases slightly with each reduction of $k$. For MARINE only $\delta = 1.1$ results in a decreased SDA. The computational amount could be decreased by up to $31\,\%$ for MEGACITY and $26\,\%$ for MARINE, with $\delta = 1.3$.

correlation between areas where the calculated growth factor exceeded $q_{max}$ and areas where the local error got very small (compare Fig. 1). Model runs with $q_{max}$ values between six and thirty for $q_{max}$ were made, but the results showed that nearly no reduction in number of function evaluations was achieved, the best was a reduction of 3 % but in most cases much less. This was most likely the case because this increase only saved a few evaluations per peak. Considering that there were only four

peaks within the first 250 steps, the benefit was small. The nearly constant overestimation of the local error seems to impact the efficiency of the solver more than the significant drops within the local error. The parameter $q_{max}$ is not able to compensate for the nearly constant overestimation of the local error because it only influences a small part of the step sizes. Thus, parameters that have impact on the majority of steps, like the safety factor, can achieve better improvements and indirectly reduce the drops of the local error.

Changes of the lower growth limit factor $q_{min}$ showed nearly no influence on the performance, mainly because the reduction factor $r$ is the key parameter for step size reduction. Compared to the known default step size control proposed by Hairer et al. (1993), the reduction factor $r$ is an addition in KPP. It helps decreasing the step size faster in the very stiff scenarios we are working with. When the step size gets rejected more than twice in a row then the next rejected step size gets multiplied with this factor, by default it is $r = 0.1$. By looking at Fig. 2 one could assume that a smaller reduction factor may help to decrease

the step size faster in the areas where it drops off drastically. Tests showed that this is true, but the difference was only marginal. Nevertheless, we found that in combination with a larger safety factor $\delta$, the reduction factor may even be increased. In the MARINE scenario this led to an additional $10\,\%$ reduction in function evaluations. However, in the MEGACITY and FOREST scenarios this also had no meaningful impact. The tests showed that there were no consistent performance changes into one direction for increases or decreases of the reduction factor, but overall values in the range $r \in [0.05, 0.2]$ performed good.



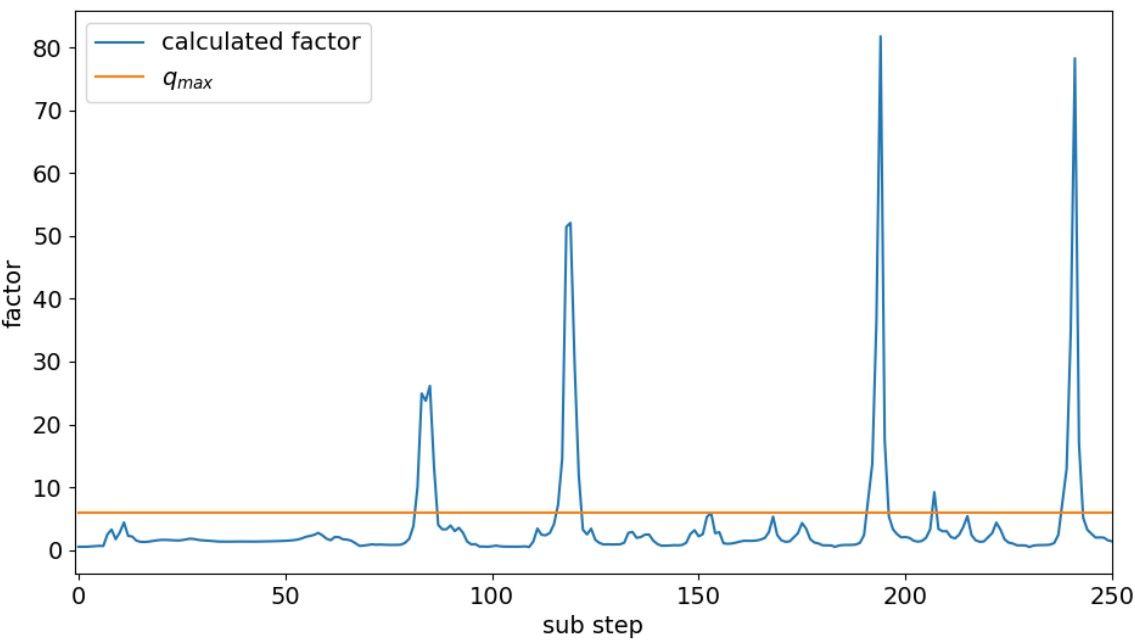

**Figure 4.** Visualization of the calculated growth factor (blue) that gets limited by $q_{max}$ (orange). The limit only gets exceeded for a few sub steps but if so, the value gets much higher, up to 80. Exceeding the limit strongly correlates with the strong drops of the local error.

Lastly, we did some rudimentary tests with the starting step size $h_{start}$. It gets used at the beginning of each model step, so it can have a meaningful impact on the efficiency of the integration even tough it is not part of the step size controller itself. Figure 2 shows that in the first model time step, the current value of $10^{-5}$ was too high, because the controller started with multiple rejections. However, for the following model steps the step size always grew at the beginning, thus higher values also seemed to be reasonable. Because of that, we tested lower and higher values, namely: $10^{-9}, 10^{-8}, 10^{-7}, 10^{-6}, 10^{-4}, 10^{-3}$. Larger values resulted in unstable model runs leading to crashes. On the other hand, smaller values yielded a slight increase in the number of function evaluations. So in short, changing this parameter did not help to increase the efficiency in our cases. Overall, the default value of $10^{-5}$ showed the best results, although better results may be achieved with an automatic starting step size selection (Watts, 1983).

### 3.2 Second-order controller

The adjustment of multiple heuristic parameters of the standard step size controller by Hairer et al. (1993) gave promising results. However, this makes adjusting the controller's behavior rather inconvenient and too specific for the ODE system at hand. In the search of a more general and flexible step size control we have implemented the H211b controller by Söderlind





(2003) from Eq. (6) and tested it with the same CAABA scenarios. The pseudocode of the implementation is shown in Appendix A.

We varied $k$ over the range of $[1.5, 3]$ which equals the lower bound of $\frac{\hat{p}+1}{2}$ and the default root exponent of $\hat{p}+1$. Since higher values of parameter $b$ create a smoother and more robust step size sequence, which somewhat stands in contrast to a most efficient step size sequence, we decided to use the lowest possible value $b = 1$ for our tests. Simulations with higher values for $b$ always resulted in more function evaluations. Even though Söderlind (2003) recommended $b > 1$, we did not encounter any issues with $b = 1$. However, this should be kept in mind if stability issues arise in other cases.

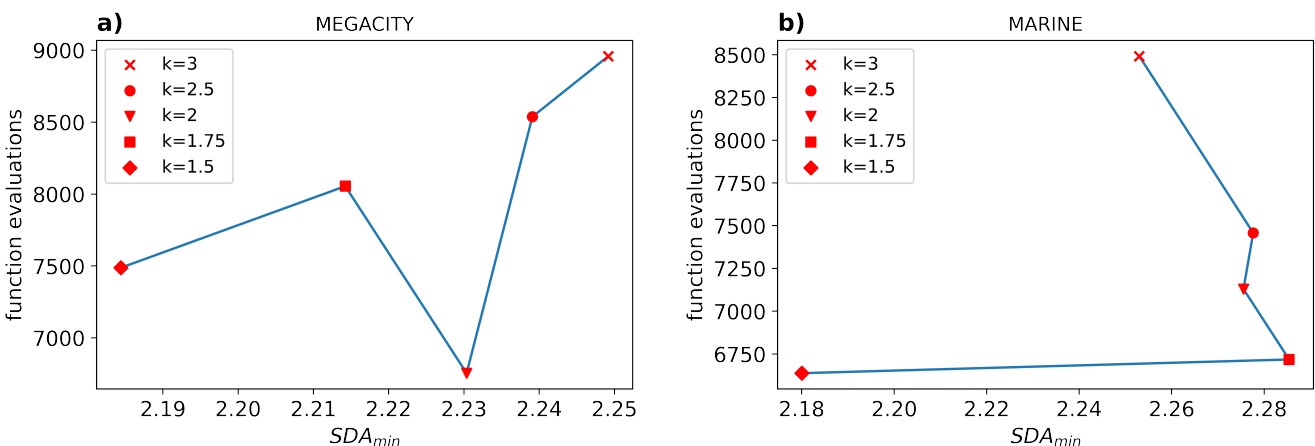

**Figure 5.** Work-precision diagrams for **a)** the MEGACITY scenario and **b)** the MARINE scenario. Generally, lower values for $k$ produce less function calls. For MEGACITY the SDA decreases slightly with each reduction of $k$. For MARINE only $k = 1.5$ results in a decreased SDA. The computational amount could be decreased by up to $31.7\,\%$ for MEGACITY and $24.1\,\%$ for MARINE, with $k = 2$ and $k = 1.75$ respectively.

Figure 5 shows that with decreasing values of $k$, the number of function evaluations dropped quite significantly in the MEGACITY and MARINE scenario, respectively. These reductions were about the same as with the increased safety factor in the default step size controller (cf. Fig. 3). Furthermore, the FOREST scenario also showed a meaningful reduction in the number of function evaluations, even though it is not as strong as in the two scenarios displayed. The MARINE scenario showed that setting $k$ to the lowest possible value (1.5) might not be optimal because the error grew stronger compared to the

saving. For the FOREST scenario, this value even slightly increased the costs. This leads to the conclusion that slightly higher values, around $k \approx \frac{2}{3}(\hat{p}+1)$ seem more appropriate as a rule of thumb.

## 4  Step size control in the global model

We applied the findings of the CAABA simulations to the EMAC model and investigated if they yielded similar reductions in the number of required function evaluations. Here, we present the results of the one-day global simulations with the setup





described in Sect. 2.4. The controller parameters were changed for each submodel independently and each simulation included only one parameter change.

## 4.1 First-order controller

Starting with simulation parameters for MECCA, it could be observed that changes of the reduction factor and the starting step size did not result in less function evaluations. Modifications of all other parameters provided improvements. The changes
of the safety factor resulted in up to $10\%$ less function evaluations, which is less than in comparison to the box model. Interestingly, the upper growth boundary $q_{max}$ showed even more improvement with a reduction of about $17\%$. In the box model, this parameter had nearly no impact. Overall, the efficiency gains were smaller than in the box model, but still have a high potential to reduce the computational burden of long global simulations. Looking into the simulations with changed parameters for SCAV, the results did not differ too much from the MECCA results. Again, the reduction factor and the starting
step size yielded no improvements, but here also $q_{max}$ showed no decrease. The different safety factors resulted in reductions between $13\%$ and $20\%$, so slightly better than in MECCA. Lastly, we changed the parameters for AERCHEM. The results for the default controller showed nearly no improvements, the only change that resulted in a small decrease of function evaluations of $3\%$ was the starting step size of $10^{-4}$. In contrast to the box model results, changing the safety factor increased the number of function evaluations. For the largest value, the increase was quite significant. Reasons might be the lower tolerance used by
AERCHEM ($10^{-3}$ instead of $10^{-2}$) and stronger chemical perturbations compared to the box model. A more detailed overview over the tests with the first-order controller can be seen in Table 3. Table C1 in the Appendix C shows the workload for each ODE system.

**Table 3.** Overview of EMAC results for different first-order controller parameter values derived from the investigations for the box model. Concrete percentages are only displayed for meaningful reductions in the number of function evaluations. The percentage numbers refer to the reduction within the corresponding ODE system, not to the whole accumulated number of function evaluations. + indicates an increase and $\approx$ represents nearly no changes. Good improvements were achieved by increasing $q_{max}$ and $\delta$ for MECCA and SCAV, but not for AERCHEM. The values in the brackets represent the default value of the parameter.

| | **Safety Factor** $\delta$ (0.9) | | | **Growth limit** $q_{max}$ (6) | | **Reduction Factor** $r$ (0.1) | | $\mathbf{H}_{start}$ ($10^{-5}$) | |
| | 1.2 | 1.5 | 1.7 | 50 | 100 | 0.05 | 0.2 | $10^{-6}$ | $10^{-4}$ |
|---|---|---|---|---|---|---|---|---|---|
| **MECCA** | $-6\%$ | $-9\%$ | $-10\%$ | $-16\%$ | $-17\%$ | $\approx$ | $\approx$ | $+$ | $+$ |
| **SCAV** | $-13\%$ | $-19\%$ | $-20\%$ | $\approx$ | $\approx$ | $+$ | $\approx$ | $+$ | $+$ |
| **AERCHEM** | $+$ | $+$ | $+$ | $\approx$ | $\approx$ | $\approx$ | $\approx$ | $+$ | $-3\%$ |

## 4.2 Second-order controller

We performed the same one-day simulations with the second-order controller. We tested $k = 2$ and $k = 1.7$, because these
values cover the range where the best reductions in the box model investigations were found. A detailed overview of the





impacts is presented in Table 4. For MECCA, up to $43\,\%$ less function evaluations were achieved, which was much better than the reduction of the default controller. For SCAV, the reduction with the second-order controller was also higher than the first-order controller with a $27\,\%$ reduction. Finally, for AERCHEM it can be seen that with the H211b controller a reduction of $13\,\%$ was possible with $b = 1$ and $k = 1.7$. In contrast to the first-order controller, the second-order controller showed

significant improvements in this case. Generally, the comparison between the columns of Table 4 shows that a smaller value of $k$ also performed better throughout for the global model simulations.

In conclusion, we were also able to reduce the number of required function evaluations in EMAC simulations with the help of the CAABA investigations. However, in the global model simulations the H211b controller was superior to the default controller, whereas for the box model the performance gains were similar.

**Table 4.** Overview of EMAC results with the H211b controller, similar to Table 3. Overall, the H211b controller performed much better than the current controller and had the only meaningful improvement for AERCHEM.

|  | **H211b** ($b = 1$) | |
|---|---|---|
|  | $k = 2$ | $k = 1.7$ |
| **MECCA** | $-38\,\%$ | $-43\,\%$ |
| **SCAV** | $-22\,\%$ | $-27\,\%$ |
| **AERCHEM** | $-8\,\%$ | $-13\,\%$ |

## 340  5   Long-term global simulations

Typically, global atmospheric simulations cover a period much larger than a single day. Therefore, there is a need to assess the accuracy of the results for much longer integration times. We hence made three one-year simulations as described in Sect. 2.4. We will first highlight how much run time can be saved with better parameters for the first-order controller and the second-order controller compared to the currently used setup over the longer simulation period. Secondly, we will show that this was

achieved with a minimal loss in quality of the integration results.

### 5.1   Run time comparison

Table 5 gives an overview of the required computation time for each simulation and the reduction that could be achieved with the new changes. The results show that with the simple parameter changes of the current step size controller, a run time reduction of roughly 1500 core-hours was achieved compared to the reference, which equals a reduction of $4\,\%$. The

reference simulations used the default parameters presented in section 3.1.2 for each submodel, while in the improved first order controller $q_{max} = 100$ was used for MECCA, $\delta = 1.5$ for SCAV and $h_{start} = 10^{-4}$ for AERCHEM. With the newly tested H211b controller, the reduction was about 3 times as high. The corresponding simulation required over 4000 core-hours less than for the reference one, about $11.4\,\%$. In addition to the run time, the last table column also shows the estimated overall





reduction for the number of function evaluations, which was nearly $20\%$. Furthermore, it should be noted that the percentage
run time reduction will always be below the function calls reduction because the model calculates many processes that are not
influenced by our changes like I/O or atmospheric physics.

**Table 5.** Comparison of runtimes for one-year EMAC simulations with the reference and two optimized controllers. Reference is the first-order controller with default parameters. Improved first-order controller has the following modified parameters: $q_{max} = 100$ was used for MECCA, $\delta = 1.5$ for SCAV and $h_{start} = 10^{-4}$ for AERCHEM. Second order controller with $b = 1, k = 1.7$ for MECCA, SCAV and AERCHEM.

|  | Run time (core-hours) | Run time reduction |
| --- | --- | --- |
| **Reference** (default parameters) | 36 968 | - |
| **Improved first-order controller** | 35 501 | $4\%$ |
| **Second-order controller** | 32 743 | $11.4\%$ |

## 5.2 Error analysis

It is important to investigate that we do not have a meaningful reduction in the accuracy of the calculated chemical abundances.
In this respect, we want to stress that not a single change was made which would influence the precision control behavior of
the integrator. The controller changes only influenced the calculation of the step size candidate to use for the next step. The
estimation of the local error and the verification that the last calculated step fits the desired tolerance, were not changed in any
way. Nonetheless, given the complexity of the chemical ODE systems and all the uncertainties in such models, we made a few
investigations of the result quality.

Given the superiority of the H211b controller, the following results only refer to a comparison between the one-year sim-
ulation with this new controller and the reference simulation. Please note that this comparison does not reflect an precision
comparison as we had for CAABA, where we used a Radau integrator with a much stricter tolerance as reference. A global
simulation with the Radau integrator would simply be not feasible with our setup for this time span.

Figure 6 shows the absolute and relative difference for the main atmospheric oxidants $O_3$, OH and $NO_3$ in the gas phase.
All plots show ground level, instantaneous values from the last model time step. For ozone, it can be seen that the relative
difference was well below $1\%$ across the whole globe and thus smaller than the used relative tolerance.

For OH, the relative difference was overall still below $1\%$, but some areas had higher values than others. For example, above
northern Africa the differences were all comparatively high, a handful of boxes even slightly exceeded $1\%$. However, this is
in no way significant, because they mainly occurred in nighttime areas where OH was very close to zero. Because of this, the
absolute difference was also very small, which makes the relative error very sensitive to small changes.



**Figure 6.** Instantaneous mixing ratio differences between the H211b and the reference one-year simulations for the last simulation day (31 December 2009, 04:00:00 UTC). The shaded area represents night. **a** Absolute difference for $O_3$. **b** Relative difference for $O_3$. All values are far below $1\%$, so below the used relative tolerance. **c** Absolute difference for OH. **d** Relative difference for OH. Nearly all value are below $1\%$. A few boxes slightly exceed this value, but only in nighttime areas with small absolute values. **e** Absolute difference for $NO_3$. **f** Relative difference for $NO_3$. Again, most values are below $1\%$, but some larger areas exceed this value. There, a few boxes go up to $20\%$.





For $NO_3$, the relative error was below $1\%$ in most areas. However, there were a few regions where it exceeded $1\%$, notably North Africa. Most of these values were only slightly higher, but a handful of outliers were above $10\%$ or even $20\%$. For earlier time steps, similar relative differences were produced in this areas, especially during nighttime. This indicates that the observed difference was not caused by error propagation. Following onto that, Figure 7 shows a time series of the average relative differences over all grid boxes of the tropospheric levels. It can be seen that on average the values were always below the

desired $1\%$ tolerance. Furthermore, the differences proceeded rather consistent and did not grow over time. This also supports that the quality of the results did not decrease. Larger deviations for $NO_3$ are not surprising, given that the $NO_3$ chemistry is coupled directly to extremely fast phase-transfer and dissociation reactions of $N_2O_5$ in cloud droplets and aerosols.

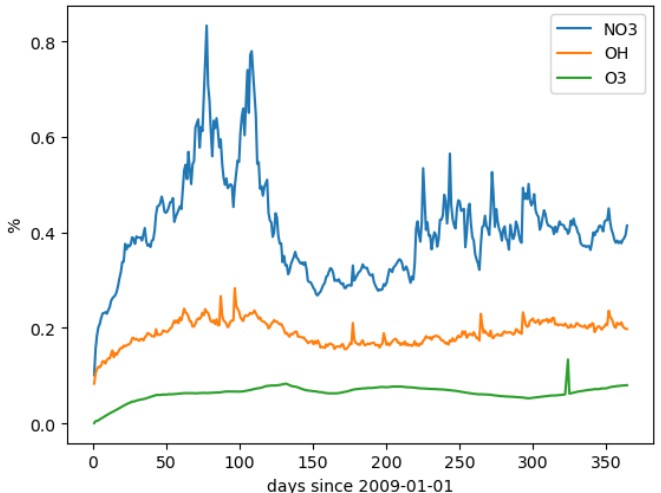

**Figure 7.** Time series of the tropospheric globally averaged relative differences of $NO_3, OH$ and $O_3$ between the reference and the H211b global one-year simulations.

Because the used mechanism simulates more than just gas-phase chemistry, we also looked into the cloud and aerosol-phase mixing ratios, focusing on OH. Figure 8 shows absolute and relative differences for the cloud-phase ($OH_l$) and aerosol-phase

in the accumulation soluble mode ($OH_{as}$) and coarse soluble mode ($OH_{cs}$). The mixing ratios in these phases were generally much lower and were distributed on a wider range of values than in the gas phase. This made the error investigation and interpretation of the results more complex. This can especially be seen for the cloud-phase chemistry. The relative difference was extremely high for the majority of the grid boxes. The reason being the very small mixing ratios and the corresponding very small absolute differences, down to $10^{-44}$. For single-precision, values below $10^{-38}$ are represented as *subnormal numbers*

which have a significantly reduced precision (noa, 2019). In areas with higher mixing ratios, colored in yellow and green, the relative difference was below $1\%$. The plots contain many empty areas because there were no clouds to calculate cloud-phase chemistry. The aerosol-chemistry showed overall higher relative differences compared to the gas phase, which is why we changed the displayed range to $10\%$. A few boxes also exceed this value, however the vast majority is still below $10\%$ and over $75\%$ of the boxes are even below $1\%$ relative error.



**Figure 8.** Instantaneous mixing ratio differences between the H211b and the reference one-year simulations for the last simulation day (31 December 2009, 04:00:00 UTC). The shaded area represents night. Absolute differences (**a**, **c** and **f**) and relative differences (**b**, **d** and **e**) for OH in the accumulation aerosol mode, coarse aerosol mode and in cloud droplets, respectively. Higher mixing ratios have a relative difference below 1%. For very small mixing ratios, the high relative difference is a result of single-precision limitations.





## 6 Load imbalance

One aspect that differs from the CAABA simulations was that EMAC simulations heavily utilized parallelization by distributing the grid cells onto multiple CPUs and their cores. For the box model, we had no parallelization. This made the number of function evaluations a perfect metric which solely represents the workload of the chemical ODE system. In the case of the global model, the parallelization could give the additional issue of unequal distribution of workload (load imbalance). With the new changes, we observed that while most areas require less function evaluations, there are very few boxes where the number of function evaluations increased. In the worst case, this could lead to an overall increase in load imbalance and with that computation time, if a single CPU core gets all boxes where the amount increases, as all other cores would need to wait, without any overall benefit. Nevertheless, as seen in the run time reductions from Sec. 5, this problem did not occur in our simulations, or at least not to a meaningful extent. However, it could occur for a different setup and should be kept in mind. As shown in Fig. 9, an increase in the number of function evaluations (the corresponding variable is called *nfun*) only happened in areas where the day-night transitions occurred. The grid decomposition algorithm does distribute these grid boxes with high workload evenly across the cores. The unintended worsening of the load imbalance is likely associated with the rapid change of chemical regime from $NO_3-$ to OH-dominated chemistry. $NO_3$ chemistry is tightly coupled to $N_2O_5$, which is very reactive in both the gas and the aqueous phase leading to enhanced ODE stiffness. This issue is already known. At dusk and dawn the strong perturbation to the chemical regime via changing photolysis rates enhances the ODE stiffness and the chemical solver needs the largest number of substeps for the integration (Christou et al., 2016). Nevertheless, porting the chemical solver to GPUs makes the load imbalance a minor issue because of the much higher degree of parallelization. The integration of the chemical ODE system can be ported to GPUs with a Fortran-to-CUDA parser (Alvanos and Christoudias, 2017; Christoudias et al., 2021).



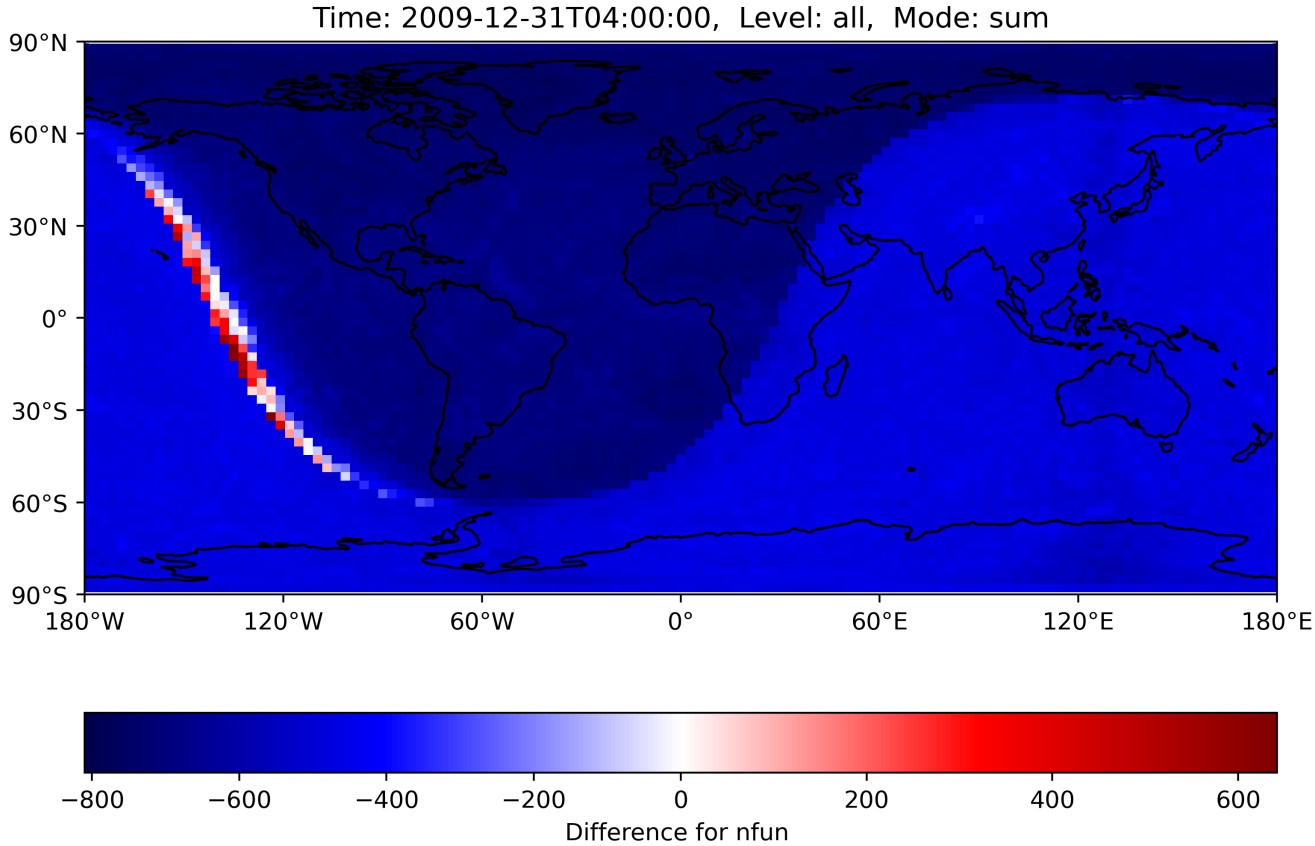

**Figure 9.** The number of function evaluations for the MECCA ODE system are summed up across all levels for the reference and the H211b simulation, afterward the difference was calculated. The dominating negative values show the decrease with the new controller. The few red boxes mark an increase in this area. Load imbalance could increase in the worst case if many red boxes are calculated by the same CPU-core. But in practice there is no meaningful impact. GPU usage should further decrease this issue. The shaded area represents night.

## 7 Conclusions

This work highlighted that the commonly used default step size controller for Rosenbrock solvers posed many inefficiencies in the context of atmospheric chemical ODE systems. For box model simulations with CAABA, we found a near constant overestimation of the local error, as well as regular drops to very small values. Both indicated the possibility for improvements by using a more efficient step size selection. Based on this, we also showed that we were able to decrease the work load of the ODE solver by simply changing the heuristic parameters of the controller. Furthermore, for the global model EMAC even better results were achieved with the new H211b controller, there the computation time of a simulation with multiphase chemistry was reduced by $11.4\,\%$. Lastly, we also took care that the results of the solver continue to be trustworthy in terms of accuracy.



Given the good performance, we recommend this controller as the new default choice. Generally, it could be worth to test step size controllers of higher order by Söderlind (2003) for further speedups of atmospheric chemistry simulations.

Additional efficiency gains might be achieved by applying the Quasi-Steady-State-Approximation (Turanyi et al., 1993, QSSA) for stiffness reduction. This option is available in our modified Rosenbrock integrator as well and allows for dynamical choice of the initial step size. However, this option is recommended only for gas-phase chemistry calculations and its impact on accuracy and efficiency still needs to be assessed.

*Code and data availability.* The CAABA/MECCA model code is available as a community model published under the GNU General Public License (http://www.gnu.org/copyleft/gpl.html, last access: 02 September 2024). The model code can be found in the Supplement. We have made the H211b controller available in EMAC version 2.55.0 and also in KPP3 branch `feature/h211b`. The Rosenbrock integrator used in this study, `rosenbrock_posdef_h211b_qssa.f90`, is available in CAABA/MECCA version 4.5.4 and MESSy version 2.55.0. The same integrator is also available in KPP3 at `rosenbrock_posdef_h211b_qssa.f90`. MESSy is licensed to all affiliates of institutions which are members of the MESSy Consortium. Institutions can become a member of the MESSy Consortium by signing the "MESSy Memorandum of Understanding". More information can be found on the MESSy Consortium website: http://www.messy-interface.org (last access: 02 September 2024). The zip-File containing the code to reproduce the global modelling results of this study is archived with a restricted-access DOI (https://doi.org/10.5281/zenodo.13768443, The MESSy Consortium, 2024). The file contains the code of the main development branch of MESSy and a patch-file containing the AERCHEM-H211b namelist setup. A cleaned-up version of the modifications will appear in the main development branch. The data produced in this study with the CAABA/MECCA box model are available at https://doi.org/10.5281/zenodo.13828706.

## Appendix A: Controller implementations

Below, we present pseudocode of our implementations of the standard first-order controller and the second-order H211b controller. Our default controller implementation only differs in two minor aspects compared to the one in Hairer et al. (1993). Given that the error tolerances are part of the local error estimation (see Eq. 5), the value of $\varepsilon$ in the numerator has to be set to one. Furthermore, we highlighted the use of a reduction factor that comes in place when a step size gets rejected twice or more in a row.

For the H211b controller we also make use of the reduction factor. Besides initializing two new variables, we only need to change the line of code that calculates the growth factor. The factor calculation is nearly identical to the Eq. 6, we only have two minor modifications. Instead of storing the last step size, the quotient $\frac{h_i}{h_{i-1}}$, which equals the last growth factor, gets stored in a new variable. In the numerator, $\varepsilon$ is again set to one for the same reason as above.



---

**Algorithm 1:** Pseudocode of the default first-order controller implementation

```
// ...
```
**while** $T < T_{end}$ **do**

 ```
 // ... calculate solution yᵢ₊₁ ...
 ```
 $err = ||l_{i+1}||;$
 $fac = \min\left(q_{max}, \max\left(q_{min}, \delta\left(\frac{1}{err}\right)^{1/(\hat{p}+1)}\right)\right);$
 $h_{new} = h \cdot fac;$
 ```
 // ... check if yᵢ₊₁ can be accepted or not ...
 ```
 **if** *rejected 2 or more steps in a row* **then**
  $h_{new} = h_{new} \cdot reductionFac;$
 **end**
 ```
 // ...
 ```
**end**

---

**Algorithm 2:** Pseudocode of the H211b controller implementation

```
// initialize two new variables
```
$facOld = 1;$
$errOld = 1;$
```
// ...
```
**while** $T < T_{end}$ **do**

 ```
 // ... calculate solution yᵢ₊₁ ...
 ```
 $err = ||l_{i+1}||;$
 $fac = \left(\frac{1}{err}\right)^{1/(b \cdot k)} \cdot \left(\frac{1}{errOld}\right)^{1/(b \cdot k)} \cdot facOld^{-1/b};$
 $facOld = fac;$
 $errOld = err;$
 $h_{new} = h \cdot fac;$
 ```
 // ... check if yᵢ₊₁ can be accepted or not ...
 ```
 **if** *rejected 2 or more steps in a row* **then**
  $h_{new} = h_{new} \cdot reductionFac;$
 **end**
 ```
 // ...
 ```
**end**

---





## Appendix B: Single Digit Accuracy (SDA)

As a measure of accuracy, we used the single digit accuracy (SDA) similar to Sandu et al. (1997b). It expresses the number of significant digits and can be calculated the following way:

$$SDA = -log_{10}\left(relativeError\right) \tag{B1}$$

To calculate the relative error of the CAABA results, we took the reference values from a fully implicit Radau-5 integrator with a relative tolerance of $10^{-7}$.

We calculated the SDA of the mean, median and largest relative error over a selection of 86 components. The aspired mean should be around a value of two, but the mean and median accuracies were much better, as shown in Table B1. In the work-precision diagrams we used $SDA_{min}$ which reflects the least precise component with the largest relative error. Only the FOREST scenario had an $SDA_{min}$ below two, all other values were noticeably above.

**Table B1.** Median, mean and lowest single digit accuracy for each box model scenario

| Scenario | $SDA_{median}$ | $SDA_{mean}$ | $SDA_{min}$ |
|----------|----------------|--------------|-------------|
| **MARINE** | 4.0688 | 3.1153 | 2.2506 |
| **MEGACITY** | 3.7607 | 3.0697 | 2.2678 |
| **FOREST** | 2.7810 | 2.4362 | 1.6478 |

## 465    Appendix C: Work load of each ODE system in EMAC

Table C1 shows the number of function evaluations needed for each system for the first simulation day of the global reference run. AERCHEM needs the least number of function evaluations with around 373 million. Followed by SCAV with nearly 480 million. Lastly, MECCA which makes up for more than half of the overall function evaluations.

**Table C1.** Comparison between the amount of work of the three ODE systems in our global model setup.

| | Number of function evaluations (first day) | Relative proportion |
|---|---|---|
| **MECCA** | 1 043 786 432 | 58.6% |
| **SCAV** | 481 043 572 | 27.0% |
| **AERCHEM** | 257 135 952 | 14.4% |





*Author contributions.* **R. Dreger**: Conceptualization, Formal analysis, Software, Visualization, Writing - original draft preparation; **T. Kir-**
**fel**: Conceptualization, Software, Writing - review & editing; **A. Pozzer**: Software, Writing - review & editing; **S. Rosanka**: Software,
Writing - review & editing; **R. Sander**: Software, Writing - review & editing; **D. Taraborrelli**: Conceptualization, Methodology, Software,
Writing - review & editing.

*Competing interests.* The co-author R. Sander is executive editor of GMD.

*Acknowledgements.* We thank Prof. Gustaf Söderlind (University of Lund) for his advice in the implementation of the H211b time step
controller. We thank Prof. Dr. Matthias Grajewski (FH Aachen) for the helpful discussions on the local error. We thank the Federal Min-
istry of Education and Research in Germany (BMBF) through the research program "MiKlip" (FKZ: 01 LP 1128A). The authors gratefully
acknowledge the Gauss Centre for Supercomputing e.V. (www.gauss-centre.eu) for funding this project by providing computing time on
the GCS Supercomputer JUWELS (Alvarez, 2021) and by the John von Neumann Institute for Computing (NIC) and provided on the su-
percomputer JURECA-DC (Thörnig, 2021) at Jülich Supercomputing Centre (JSC). The authors gratefully acknowledge the Earth System
Modelling Project (ESM) for funding this work by providing computing time on the ESM partition of the supercomputer JUWELS at the
Jülich Supercomputing Centre (JSC).



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
