# Peer review of "Optimized step size control within the Rosenbrock solvers for stiff chemical ODE systems in KPP version 2.2.3\_rs4"

_Geoscientific Model Development, 2024_

## Author Comment (AC1)

**gmd-2024-166 – Reply to referees**

We thank the referees for their constructive comments. In the course of the open discussion we have become aware of a minor bug in the diagnostics of KPP substeps for the aqueous-phase chemistry in aerosols as implemented in AERCHEM. However, only the absolute numbers change and the relative change is almost identical. We have changed Table C1 with the correct values.

In the following, we have listed the comments in normal font color and style, followed by our replies in blue and modifications to the text in red.

**Referee #1**

*Dreger et al. present an optimized first-order step size controller for the Rosenbrock solver within the Kinetic Pre-Processor (KPP) package, which is often used for solving stiff systems of ODEs used in atmospheric chemistry. The updated step size controller achieves double-digit improvements in computational performance in gas-phase and complex aqueous-phase chemistry with minimal (<1%) errors for main tropospheric oxidants in a 1-year simulation period. This work is important to the development of chemical solvers in atmospheric chemistry; it tackles a new aspect of improvement in contrast to previous efforts in reducing the ODE system size, improved solvers, and use of GPUs, by improving the step size controller, and implements the work in a widely used KPP software package making it adoptable by other atmospheric chemistry models immediately. The work is well written and in scope for GMD, and I recommend its publication. I have a few questions for clarification on some aspects of the work presented by the authors.*

We thank Referee #1 for the very positive review. The questions help us improving the clarity of the manuscript. We appreciate it.

*1) L33: "atmospheric chemical kinetic models satisfy this criterion when the mechanism contains very fast as well as very slow reactions..." I would rephrase to emphasize that atmospheric chemistry inherently has this property because of the large range of time-scales involved in a typical atmosphere - "when the mechanism contains" implies to me that one can select a mechanism that doesn't have this property, but this wouldn't be representative of the typical (Earth) atmosphere.*

We agree with this suggestion and will change it accordingly.

2) Figure 2 notes that the first model step requires significantly more substeps than the others. I understand this may be because the initial condition provided to the solver is a condition that was perturbed by other non-chemistry operators (e.g., advection, emissions) and the solver needs to first adapt to this new condition. I see that in L280-289 the authors have experimented with changing this initial step size. I am curious about this further optimization - could the authors elaborate on "for the following model steps the step size always grew at the beginning" - is this only observed in the box model environment? Are there other operations on the chemical concentrations between integrator time steps in the box model? I wonder if such an optimization could be more advantageous in a global model where operator splitting would have a greater effect on the solver's initial time step.

The reviewer is correct in stating that non-chemistry operators (emission and deposition) disturb the chemical conditions during the box model run between two calls of the KPP solver. However, this is not what we refer to with the expression "first model step" in the caption of Fig. 2. Instead, with "first model step" we refer to the very beginning of the model run. Only at this point are the concentrations so far away from steady-state that $h_{start}$ had to be reduced by the ODE solver.

3) Eq. 7 - is "equal with exclamation mark" here denoting an estimation or a note to the reader?

The equal sign with exclamation mark means "should be equal to". As this is mathematically correct, we did not change the text.

4) Table 3 - could the authors clarify the difference between "the reduction within the corresponding ODE system" vs. "the whole accumulated number of function evaluations"? If it's not referring to the function evaluations, is this reduction a reduction in wall-clock time?

If we assume that the ODE system of MECCA needs 100 function evaluations, the system of SCAV needs 50 and the one of AERCHEM needs 200. After a change to the controller of MECCA the number of function evaluations of MECCA reduces to 80. Then "the reduction within the corresponding ODE system" means a 20% reduction (in this case for MECCA), but compared to "the whole accumulated number of function evaluations" (to the total) it would only be a 5,7% reduction. We just wanted to highlight that the reduction values are always relative to the ODE system where the change was made.

5) In Table 5 I would note the % of function calls reduction as well which I believe is a better representation of the improvement in the solver itself (for the same arguments made in the manuscript that the model has other processes like I/O which also take significant amount of computational time), even if run time reduction is most relevant to model users.

We will add a column with the reduction of number of function evaluations to Table 5.

6) Section 5.2 Error analysis Figure 7 - the authors show main tropospheric oxidants (OH, O3, and NO3) percentage difference compared to a simulation without the improved step size controller. I was wondering if a plot including all the species in a lighter color in the background (similar to Figure 7 in Shen et al., 2022 - https://doi.org/10.5194/gmd-15-1677-2022) would help visualize the effect on other species and identify potential outliers in relative differences.

We replaced Figure 7 with four plots similar to the referenced one in Shen et al. 2022. The plots are divided into the species of MECCA, SCAV, accumulation soluble mode from AERCHEM and coarse soluble mode from AERCHEM. The plots are also shown below.

[Figure]

7) Figure 8 d) and f) are quite saturated in color scale; as the authors have noted this may be just numerical noise because of
concentrations that are too small. Maybe the same figure could be made to filter out (maybe mark as grey) small concentrations
(by some percentile or by the single-precision limit?) so the % difference on grid boxes with "meaningful" (non-numerical
noise) values can be seen better?

We thank the referee, as this suggestion would definitely increase the readability of Figure 8f). We have therefore changed
it accordingly by filtering out concentrations below a threshold of 1e-32. The figure below shows the updated version.

[Figure]

Time: 2009-12-31T04:00:00, Level: 31.0

Relative difference for OH_l in %

8) The authors included a discussion of load imbalance in Section 6, which is a known problem in atmospheric chemistry models where the terminator zone has a much greater number of internal step sizes, making parallel simulations spend wall-clock time on "waiting" (e.g. MPI barriers). Figure 9 is a good visualization, the # of step sizes has been used in many papers to show this imbalance, but in terms of visualizing improvement in this imbalance between the original and the optimized solver, I wonder if the authors would be able to plot a histogram of the distribution of the # of function evaluations in # of grid boxes, and show if the distribution of function evaluations has changed between the original and optimized solver.

Maybe there is a slight misunderstanding in this case. We included Section 6 to show that a possible downside of the new controller is a potential increase in load imbalance (line 401: "In the worst case, this could lead to an overall increase in load imbalance"). So there is no improvement in this regard. The overall reduction in the number of function calls overweights this potential increase in load imbalance. We created the suggested histogram and think that it also supports this thesis. The added figure below shows, as expected, that there are much more grid boxes with a very low number of function evaluations. But at the same time, the number of boxes with rather high number of function evaluations also increases slightly. So the distribution is more spread out. We plan to include the histogram into Section 6.

[Figure]

9) Another note on the load imbalance issue, the authors note in caption of Figure 9 that "Load imbalance could increase in the worst case if many red boxes are calculated by the same CPU-core" - I suppose this depends on the MPI/OpenMP decomposition used by the MECCA model. How does MECCA decompose the domain? Different models have different approaches, in the GEOS-Chem High Performance model (10.1029/2020MS002064 Figure D1) the domain is decomposed in a trivial way on each face of the cubed-sphere, but other models like CESM have more elaborate decompositions which aim to distribute sunlight evenly across cores (for the benefit of load balancing radiation code) which would benefit atmospheric chemistry.

The parallel decomposition depends on the dynamical core (base model) that is used in MESSy (ECHAM5 in the EMAC configuration). The model grid is split horizontally using two run-time parameters. By selecting the number of processes in the longitudinal and latitudinal directions a rectangular decomposition is obtained. In ECHAM5, two such rectangular sets of grid points, diametrically opposed wrt. the Equator, are assigned to one processor. Similarly to CESM, this is done for counteracting the load imbalance associated with radiation transfer and photochemistry (Christou et al., 2016). We added a shorter explanation by adding text at line 399:

In MESSy the parallel decomposition depends on the dynamical core that is used (ECHAM5 in the EMAC configuration). The model grid is split horizontally selecting the number of processes in lat-lon rectangular sets of grid points. Two diametrically opposed sets are assigned to one processor. This is done for counteracting the load imbalance associated with radiation transfer and photochemistry (Christou et al., 2016).

10) Do the authors plan to contribute this improved step-size controller to the mainline KPP code for use by other models by default in the Rosenbrock solver?

We already have a branch in the KPP GitHub repository for this: https://github.com/KineticPreProcessor/KPP/tree/feature/h211b. The branch will eventually be merged into main and the code will be available in the next KPP release.

**References**

Christou, M., Christoudias, T., Morillo, J., Alvarez, D., and Merx, H.: Earth system modelling on system-level heterogeneous architectures: EMAC (version 2.42) on the Dynamical Exascale Entry Platform (DEEP), Geoscientific Model Development, 9, 3483–3491, https://doi.org/10.5194/gmd-9-3483-2016, publisher: Copernicus GmbH, 2016.

---

## Author Comment (AC2)

**gmd-2024-166 – Reply to referees**

We thank the referees for their constructive comments. In the course of the open discussion we have become aware of a minor bug in the diagnostics of KPP substeps for the aqueous-phase chemistry in aerosols as implemented in AERCHEM. However, only the absolute numbers change and the relative change is almost identical. We have changed Table C1 with the correct values.

In the following, we have listed the comments in normal font color and style, followed by our replies in blue and modifications to the text in red.

**Referee #2**

*This is a nice paper examining how step size control/optimization impacts computational efficiency and accuracy when simulating atmospheric (multiphase) chemistry in box and 3D global chemical transport modeling. It is relatively clear and well-written and presents novel results that will be of significant value to atmospheric chemistry modeling community, particularly for operational 3D modeling where efficient yet accurate approaches are required. It is well-suited for GMD. I recommend publication with the consideration of minor edits/questions.*

We thank Referee #2 for the very encouraging review. Here below we address the questions.

1) Line 67. From "represents the currently used controller used in our applications that we investigate" to "represents the controller we investigate…"

We will change this accordingly.

2) Line 91. From "robuster" to "more robust"

We will change this accordingly.

3) Table 1 caption. Change "over" to "of"

We will change this accordingly.

4) Lines 127-132. Are these three scenarios specified only by changing input parameters or are there changes to the mechanism treated? (You might want to mention this here and define the changes that are made to represent the different scenarios somewhere.)

Yes, the three scenarios are among the available scenarios of the CAABA/MECCA model. They are specified by setting different input parameters and emissions. We will mention this more explicitly and add references that include the initial concentrations and emissions.

5) Line 145. Remove the word "binary"?

We will change this accordingly.

6) Line 153. Should "GMXE" be changed to "GMXe"?

45 We will change this accordingly.

7) Lines 186-187. This is unclear to me. Output is written every 23rd hour? Or every day at hour 23? How does this lead to output for every hour of the day? Do you just mean that there are outputs for each hour (0-23) as you cycle through the days of the year (writing every 23rd hour) but not outputs for every hour of each day?

50

It is every 23rd hour so that for each subsequent simulated day the output is at a local time one hour earlier. Therefore, after 24 days we would have output for each local hour (0-23) of the day. This allows to sample the model state in fairly all the chemical regimes. We will rephrase it to make it more clear.

55 8) Line 233. Typo – change "suite" to "suit"

We will change this accordingly.

9) Line 262. Remove "for qmax" (redundant)

60

We will change this accordingly.

10) Line 272. From "decreasing" to "decrease".

65 We will change this accordingly.

11) Line 279. From "good" to "well".

We will change this accordingly.

70

12) Line 281. From "tough" to "though".

We will change this accordingly.

75 13) Line 301. From "scenario, respectively." to "scenarios."

We will change this accordingly.

14) Section 4.1 and elsewhere (e.g., line 315): it might be better to say "fewer" function evaluations than "less".

80

We will change this accordingly.

15) Also, any thoughts on why the qmax changes had more influence in the global simulations wrt MECCA compared to SCAV and the box modeling where it was not very impactful?

85

The most likely reason for this is, that the MECCA chemical mechanism in the global simulations does not cover the aqueous phase chemistry, making it less stiff. Especially in the stratosphere during nighttime the solution for some model time steps only requires very few sub steps. However, with the default qmax value of 6 the step size can't increase fast enough from the small step size that is required at the beginning of the model time step. Because of this the solver will calculate much more sub

90 steps than the ODE system would need. So the default qmax value kind of implicitly introduced a lower limit of sub steps that will be calculated, which is unnecessarily high for the MECCA/EMAC case.

16) Line 326. Change "over" to "of".

We will change this accordingly.

17) Table 3 caption: Change from "qmax and [safety factor] for MECCA and SCAV" to "qmax and [safety factor] for MECCA, [safety factor] for SCAV,". Change "brackets" to "parentheses".

We will change this accordingly.

18) Line 361. Remove comma.

We will change this accordingly.

19) Line 365. From "an precision" to "a precision"

We will change this accordingly.

20) Line 377. Change "areas" to "area"

We will change this accordingly.

21) Line 380. Remove "proceeded rather consistent and"?

We will change this accordingly.

22) Line 390. What is this reference?

Thank you for pointing out a mistake here. We will change the reference to be displayed correctly.

23) Would it be worthwhile to look at the relative differences for an aerosol species that has important chemistry in cloud and aerosol water (e.g., sulfate)?

We think that this is a good suggestion. We added plots with relative differences of $SO_4^{2-}, NO_3^-, NH_4^+$ and $Cl^-$ as supplementary material and at the end of this reply. For these species the relative differences were within an acceptable range either. Accordingly, we have added the following text to line 395:

For completeness, we added the relative differences for $NO_3^-$ and $Cl^-$ (Figure S1) and for $SO_4^{2-}$ and $NH_4^+$ (Figure S2) in their respective aqueous and aerosol-phases as supplementary material.

24) Line 423: Change "worth" to "worthwhile"

We will change this accordingly.

25) Line 448. Change "...to the Eq. 6, we only have two" to "...to Eq. 6 with only two"

We will change this accordingly.

[Figure]

[Figure]

140 **References**